# Liposomal Encapsulation of Ascorbyl Palmitate: Influence on Skin Performance

**DOI:** 10.3390/pharmaceutics16070962

**Published:** 2024-07-20

**Authors:** Aleksandra Stolić Jovanović, Vanja M. Tadić, Milica Martinović, Ana Žugić, Ivana Nešić, Stevan Blagojević, Nebojša Jasnić, Tomislav Tosti

**Affiliations:** 1“Filly Farm” Pharmacy, Miloša Velikog bb, 11320 Velika Plana, Serbia; 2Department for Pharmaceutical Research and Development, Institute for Medicinal Plant Research “Dr. Josif Pančić”, Tadeuša Koscuška 1, 11000 Belgrade, Serbia; vtadic@mocbilja.rs (V.M.T.); azugic@mocbilja.rs (A.Ž.); 3Department of Pharmacy, Faculty of Medicine, University of Nis, Boulevard Dr. Zorana Djindjića 81, 18000 Nis, Serbia; milica.martinovic@medfak.ni.ac.rs (M.M.); ivana.nesic@medfak.ni.ac.rs (I.N.); 4The Institute of General and Physical Chemistry, Studentski trg 12/V, 11158 Beograd, Serbia; sblagojevic@iofh.bg.ac.rs; 5Faculty of Biology, University of Belgrade, Studentski trg 16, 11158 Beograd, Serbia; jasnicn@bio.bg.ac.rs; 6Institute of Chemistry, Technology and Metallurgy-National Institute of the Republic of Serbia, University of Belgrade, Studentski trg 12-16, 11158 Belgrade, Serbia; tosti@chem.bg.ac.rs

**Keywords:** ascorbyl palmitate, liposomes, cream, emulgel, tape stripping

## Abstract

L-ascorbic acid represents one of the most potent antioxidant, photoprotective, anti-aging, and anti-pigmentation cosmeceutical agents, with a good safety profile. However, the main challenge is the formulation of stable topical formulation products, which would optimize the penetrability of L-ascorbic acid through the skin. The aim of our research was to evaluate the performance of ascorbyl palmitate on the skin, incorporated in creams and emulgels (2%) as carriers, as well as to determine the impact of its incorporation into liposomes on the penetration profile of this ingredient. Tape stripping was used to study the penetration of ascorbyl palmitate into the stratum corneum. In addition, the sensory and textural properties of the formulations were determined. The liposomal formulations exhibited a better penetration profile (*p* < 0.05) of the active substance compared to the non-liposomal counterpart, leading to a 1.3-fold and 1.2 fold-increase in the total amount of penetrated ascorbyl palmitate in the stratum corneum for the emulgel and cream, respectively. Encapsulation of ascorbyl palmitate into liposomes led to an increase in the adhesiveness and density of the prepared cream and emulgel samples. The best spreadability and absorption during application were detected in liposomal samples. The obtained results confirmed that liposomal encapsulation of ascorbyl palmitate improved dermal penetration for both the cream and emulgel formulations.

## 1. Introduction

L-ascorbic acid and its derivatives (ascorbyl palmitate, magnesium ascorbyl phosphate, tetraisopalmitoyl ascorbic acid) are basic ingredients in anti-aging products due to their potent antioxidant activities and photoprotective properties and their involvement in collagen biosynthesis [1,2,3,4,5,6]. By interfering with tyrosinase, L-ascorbic acid can treat hyperpigmentation, melasma, and sunspots [7,8]. Also, it was successfully applied for reducing post-laser resurfacing erythema and decreasing acne scars [9]. Some clinical studies suggest that ascorbyl palmitate possesses anti-inflammatory activity and exhibits possible beneficial effects in treating some inflammatory dermatoses [8,10].

The topical application of L-ascorbic acid brings different challenges (instability, low skin penetration, easy oxidation, etc.) [11]. Thus, there is a need for the development of more stable derivatives and new safer delivery systems to achieve the desired efficiency in different skin conditions [12,13,14,15].

Emulsion-based formulations are commonly utilized for the topical administration of active ingredients due to their ability to effectively interact with both lipids and water in the outermost layer of the skin, known as the stratum corneum [16]. Consequently, creams formulated as emulsions, particularly oil-in-water (O/W) creams, have gained significant popularity. They are easily applied, spread smoothly, and impart a light sensation on the skin. These sensory properties are often critical for consumer acceptance and patient compliance and significantly influence the sales dynamics of the products [17].

The widespread use of emulgel in cosmetics and pharmaceutical preparations is due to favorable properties such as better loading capacity, selectivity to a specific site, and suitability for medications with a short biological half-life and narrow therapeutic window [18,19]. Better efficiency has been reported in terms of spreadability, adhesion, viscosity, and extrusion. These transparent, attractive, and non-greasy novel vehicles for drug delivery are obtained by the conversion of classic emulsions with the addition of a gelling agent to the water phase [20,21]. Therefore, these systems combine the advantages of both emulsions and gels. The gel matrix in the aqueous phase of the emulsion provides stabilization and controlled release of active substances, thereby reducing the likelihood of issues such as phase separation and creaming typically associated with emulsions [22].

In the present work, we focused on an amphiphilic L-ascorbic acid derivative, ascorbyl palmitate, which, compared to L-ascorbic acid, has better stability and ability to penetrate the skin [23,24]. The instability of L-ascorbic acid is mainly due to its oxidative degradation process, catalyzed by metal ions and/or by light, and significantly influenced by its initial concentration [25]. According to previous studies, some strategies can be considered to solve this issue, which include the freeze-drying method and the addition of a co-antioxidant. A novel and promising approach for enhancing ascorbyl palmitate stability involves the incorporation of additional antioxidant 4-(tridecyloxy) benzaldehyde oxime in O/W microemulsions [26]. In the study by Khan et al., ascorbyl palmitate and sodium ascorbyl phosphate were loaded in W/O/W multiple emulsions, which enabled synergistic and long-lasting effects on the skin [27]. The freeze-dried ascorbyl palmitate (AsP)-containing liposome could protect the drug and enhance its stability. This type of liposome exhibited similar skin permeation and localization properties as observed by freshly prepared liposomes [28]. Also, enhanced skin permeation of ascorbyl palmitate could be achieved by its encapsulation in liposomes which are then dispersed into the poloxamer hydrogel matrix [29]. Therefore, an appropriate carrier system, initial concentration, and storage conditions are crucial when it comes to the stability of ascorbyl palmitate in topical formulations [24].

Currently, phospholipids are very attractive and intensively investigated components in dermal products as emulsifiers, liposome-forming agents, or wetting agents [30]. Liposomes are small vesicles with a unique structure and numerous advantages that include biodegradability and nontoxicity, moisturizing and restoring action of the constitutive lipids, easiness of preparation, and continuous supply of active ingredients over a sustained period [31,32]. Not only do they act as “drug transporters” but also as “drug localizers” that can avoid systemic absorption and consequent side effects [15].

Tape stripping is simple, minimally invasive, and the most widely used technique for measuring the input kinetics and elimination of the drug from the stratum corneum [33]. In this manner, the influence of the type of formulation on the delivery and concentration of different active entities, whose target site is either in the SC or viable underlying tissues, can be determined [34,35].

The aim of our research was to examine the dermal penetration efficacy of ascorbyl palmitate from liposomal and non-liposomal creams and emulgels, incorporated in the same percentage (2%), in terms of textural characteristics and stability.

## 2. Materials and Methods

### 2.1. Materials

The following oil-phase components were used for the preparation of the creams and emulgels: Caprylic/capric triglycerides (Myritol^®^ 318) from BASF (Ludwigshafen, Germany), olive oil (Paryol 165 OL/R) from A&A FratelliParodi (Campomorone, Italy), and isopropyl myristate from Merck Schuchardt OHG (Hohenbrunn, Germany). MontanovTM82 and MontanovTM14 from Seppic (La Garenne-Colombes, France) were used as emulsifiers. As a gelling agent, hydroxyethylcellulose from Chem Point (Kraków, Poland) was used, while propylene glycol from Fagron (Rotterdam, The Netherlands) was used as a humectant and Euxyl PE 9010 (phenoxyethanol (and) ethylhexylglycerin (90:10)) from Schulke&Mayr (Norderstedt, Germany) as a preservative. The active component of the tested samples was ascorbyl palmitate (Chemkart, Mumbai, India). For obtaining liposome dispersion, Phosal 40IP (Lipoid, Steinhausen, Switzerland) was used. Purified water originated from the Faculty of Medicine (University of Niš, Serbia). L-Ascorbic acid and meta-phosphoric acid (MPA) were purchased from Sigma-Aldrich.

### 2.2. Methods

#### 2.2.1. Preparation of Liposome Dispersion

The primary liposome dispersion (Table 1) was made using a T18 basic Ultra-Turrax^®^ Homogenizer (IKA Werke, Breisgau, Germany). The final liposome dispersion was made using a LiposoFast^®^ LF-50 Avestin extruder. The dispersion was extruded twice through a 0.2 μm poly-carbon filter membrane (Whatman, Marlborough, MA, USA) and later twice through a 0.1 μm poly-carbon filter membrane (Whatman, Marlborough, MA, USA). The liposome dispersion obtained in this way was stored in a refrigerator at a temperature of 2–8 °C and used to prepare liposomal topical formulations [36].

#### 2.2.2. Preparation of the Creams and Emulgels

For the purposes of this study, two creams and two emulgels (Table 2) were made, one of each type containing a liposomal dispersion (Table 1). The creams were prepared by a standard emulsion preparation procedure, while the principle of the preparation of emulgels was in accordance with previous studies [11]. The active substance was added at the same concentration to all formulations, either as a solution of ascorbyl palmitate in isopropyl myristate (cream and emulgel) or as an ascorbyl palmitate liposomal dispersion in Phosal phospholipids (lipocream and lipoemulgel).

#### 2.2.3. Physico-Chemical Characterization of the Liposome Dispersion

The size of liposomes in the primary liposome dispersion was determined by measuring the turbidity of 0.025% of liposome dispersion diluted with phosphate buffer (pH = 7.2) using the equation developed by Ohsawa et al. [37].

Analysis of liposome size in the final liposome dispersion was carried out by using a Zetasizer (Nano series) ZS 90, Malvern Instruments, whose working principle is based on Dynamic Light Scattering (DLS) and Photon Correlation Spectroscopy (PCS) [38].

##### Evaluation of Encapsulation Efficiency

The ascorbyl palmitate encapsulation efficiency was determined by using the previously established protamine aggregation method [39]. It was applied to separate the free ascorbyl palmitate from ascorbyl palmitate entrapped within liposomes. In order to evaluate liposome entrapment efficiency, the liposomal dispersion was mixed with an equal volume of protamine solution (10 mg/mL) on a vortex mixer for 60 s. Then, the obtained mixture was centrifuged at 20,000 rpm for 15 min at 17 °C using a Hettich Universal 320R centrifuge (Andreas Hettich GmbH, Tuttlingen, Germany). An HPLC method was used to determine ascorbyl palmitate both in the supernatant and in the sediment (dissolved in 10 mL MeOH), to make sure that the entire quantity of ascorbyl palmitate used in the formulation (entrapped into liposomes and free form) was detected.

The encapsulation efficacy was calculated as the ratio between ascorbyl palmitate detected in the formulation over the initial concentration used to make the formulation.
$$\mathrm{EE}\%\;=\frac{W_{t}}{W_{i}}\times 100\%$$
where W_t_ is the total amount of ascorbyl palmitate in the liposomes and W_i_ is the total quantity of ascorbyl palmitate added initially during the preparation of the liposomal formulation.

#### 2.2.4. Physico-Chemical Characterization of Creams and Emulgels

All formulations were analyzed organoleptically (color, smell, appearance) in order to determine potential instability of the emulsion. The pH of a 10% solution of the prepared formulations was measured at room temperature (pH 211 Microprocessor pH Meter, Hanna Instruments, Woonsocket, RI, USA). The electrical conductivity of the formulations was measured by directly immersing the conductometer electrode CDM 230 (Radiometer, Copenhagen, Denmark). The same measurements were performed after centrifugation (at 3000 rpm for 15 min, at the room temperature) and after three cycles of an accelerated stability test, conducted at three different temperatures for 24 h (room temperature, temperature of 5 ± 2 °C, temperature of 45 ± 2 °C). Finally, a long-term stability test was conducted after 1 month of storage of the samples at room temperature.

#### 2.2.5. Texture Profile Analysis (TPA) of Creams and Emulgels

The CT3 Texture Analyzer (Brookfield, AMETEK Inc., Middleboro, MA, USA) was used for texture profile analysis (TPA). The experimental conditions were set as presented in Table 3.

#### 2.2.6. Examination of Sensory Properties

Twenty volunteers were involved in the sensory study where they were asked to fill in a questionnaire regarding the sensory characteristics of the formulations prior to, during, and after the application. For each sensory characteristic, a list of descriptive terms was provided. This questionnaire was formulated in accordance with our previous studies [11,40,41]. All participants were in a well-lit room, where the temperature was set at 21 ± 2 °C and a humidity of 45 ± 3%.

#### 2.2.7. Tape Stripping

Six volunteers (all women, 23 to 30 years old), with no history of dermatological disorders, participated in the study. They were asked not to apply or use any products on the left forearm one day prior to the study. A frame with square-shaped holes (2.5 cm × 2.5 cm) was used to mark the place for the application of 0.2 mL of each of the formulations (LE, E, LC, C). Before the application of the formulations, transepidermal water loss (TEWL) was measured on each site using a Tewameter^®^ TM 300 probe attached to a Multi Probe Adapter MPA^®^ 9 (Courage&Khazaka Electronic GmbH, Köln, Germany). The formulations were applied to four sites while taking care that no preparation was applied 5 cm above the wrist and 1 cm below the elbow. Two hours after the application of the samples to the intended area, by using a laboratory glove, the excess sample that was not absorbed into the skin was wiped with gauze, and tape stripping was conducted. For this procedure, Transpore 3 M adhesive tapes were cut in the square dimensions 2.5 cm × 2.5 cm (Figure 1), carefully adhered to each site, using a roller to apply a uniform pressure, and removed. For each site, 16 pieces of adhesive tape were removed and collected. TEWL was measured at each site after 4, 8, 12, and 16 strip applications with the aim of detecting the removal of the stratum corneum. The literature data suggest that an eightfold increase in TEWL indicates that the stratum corneum was completely removed [35]. The content of ascorbyl palmitate in each strip was analyzed using an HPLC method.

#### 2.2.8. HPLC Analysis

##### The Chromatographic Conditions

All analyses were performed using a Thermo Scientific (DionexUltiMate 3000) HPLC system with a binary pump and electrochemical detector with glassy carbon as the working electrode. The Acclaim Polar Advantage II C18 column 5 µm (150 × 4.6 mm) was used as the stationary phase with 20% m-phosphoric acid adjusted at pH 2.00 as the mobile phase. Instrument control and data acquisition were carried out by the Chromeleon7 Chromatography Data System (Thermo Scientific).

The method for the determination of ascorbic acid was properly validated in accordance with the Eurofins method (HPLC Method Validations—Navigating the Pitfalls, Joe Page, Eurofins Advantar 16 November 2023) and the descriptive statistics of the method, such as linearity, reproducibility, limit of detection, and limit of quantification, were determined. In addition, extraction efficiency, i.e., recovery, of the method was investigated in various matrices such as brain, liver, kidney, and skin tissues. The method is specific and selective under given chromatographic conditions [42]. In addition, selectivity was confirmed by Suw Young Ly et al. [43] who reported that this method can be used for the determination of ascorbic acid in biological and pharmaceutical matrices.

The correlation coefficient of the calibration curve was 0.9994 with a relative standard deviation of 1.4%. The obtained limit of detection was 0.0032 ppm, whereas the limit of quantification was 0.0099. The recovery was determined for all matrices and the recovery range was from 93 to 107%. The quality control mixture used for monitoring instrument performance had a concentration of 10 ppm. The peak resolution was >2.0 [44].

##### Sample Extraction

The test strips were mixed with 2 mL of mobile phase and sonicated in an ice-cold ultrasonic bath at 0 °C for 15 min. The solution was filtered through a 0.22 µm nylon syringe filter and kept in a freezer until analysis.

#### 2.2.9. Ethical Standards

All participants who were involved in the sensory study and tape stripping were well informed about the course of the study and signed informed consent. The entire study was conducted in accordance with the Helsinki Declaration and the relevant guidelines and published recommendations, while permission was provided by the Ethics Committee of the Medical Faculty in Niš (Serbia), protocol code 12-11272/2-1, from 25 September 2023.

#### 2.2.10. Statistical Analysis

The textural analysis results for each parameter were presented as mean ± standard error. A statistical analysis was performed using one-way analysis of variance (ANOVA) with a post hoc Tukey’s test by using IBM SPSS Statistics 20 (IBM Corporation, Armonk, NY, USA). Differences at *p* < 0.05 were considered statistically significant. The results of the sensory analysis were presented using an ordinary scale from 1 to 10.

## 3. Results and Discussion

L-ascorbic acid is present in the skin at relatively low amounts (∼41 ng/mg (dry weight) for the entire skin). As the stratum corneum contains only 7 ng/mg (dry weight) of L-ascorbic acid, topical application is justified in order to increase its cutaneous levels [45]. As a hydrophilic and charged molecule (pKa 4.2), one of L-ascorbic acid’s disadvantages is poor skin penetration [46]. Modification of L-ascorbic acid hydroxyl groups has an important influence on its therapeutic properties, leading to improvements in its antioxidant potential as well as its antitumor and antiviral activities [47]. The introduction of the lipophilic moieties into the structure of L-ascorbic acid increases the thermal and oxidative stability of the derivatives [48], at the same time affecting their mobility and distribution through the phospholipid bilayer membrane [49].

In recent years, there has been intensive research regarding the delivery of active components into different layers of the skin using specific types of liposomes. It has been demonstrated that cellular L-ascorbic acid intake increased significantly when yeast-based liposomes were used as a carrier system [50,51]. Moreover, according to the literature data, the encapsulation of L-ascorbic acid into a lipospheric form resulted in better transport into the deeper layers of the skin [52,53]. Serrano et al. showed in their study that a new ascorbate-phosphatidylcholine liposome formulation as a carrier system improved the topical ascorbic acid treatment of skin [54]. The literature data point out that the formulation type affects the release rate of ascorbyl palmitate from topical preparations, while several papers deal with stability studies of L-ascorbic acid when it was encapsulated into either chitosan-coated or pectin-coated liposomes [55,56,57]. In addition, it was demonstrated that the liposomal formulations for topical application significantly increased the rate and extent of L-ascorbic acid ester absorption into the epidermis [58].

In our paper, we examined four types of formulations (Table 2) with incorporated ascorbyl palmitate: cream (C), emulgel (E), lipocream (LC), and lipoemulgel (LE). In LC and LE, ascorbyl palmitate was primarily incorporated in a liposome dispersion. The assessment of the physico-chemical properties of the tested samples is presented in Table 4.

The emulgels and creams appeared to be stable according to the results obtained before and after centrifugation.

The diameter of the particles of the primary liposome dispersion, calculated spectrophotometrically, was 0.863 µm, while the diameter of particles of the final liposome dispersion was 0.783 µm. The pH of the final liposome dispersion was 4.13 and the zeta potential value was −63.67 ± 0.81 mV, while the polydispersity index (PdI) was 0.67 ± 0.01. If a PdI value is greater than 0.3, it indicates that diameters of liposomes are within a wide range [59]. Zeta potential (surface charge) is an important parameter that determines the stability of liposomal dispersions. Particles with a zeta potential of greater value than +30 mV and less than −30 mV are considered stable [60]. According to the obtained zeta potential value, the prepared liposomal dispersion was stable. Hence, in spite of the wide range of the particle size, our result was in the line with our goal to encapsulate ascorbyl palmitate and form stable liposomes. The entrapment efficiency of ascorbyl palmitate into formulated liposomes was also evaluated. The average entrapment efficiency of ascorbyl palmitate-loaded liposomes was 92.02%, indicating the successful encapsulation of the active substance into liposomes.

In the present study, we provide new evidence on the topical bioavailability of novel ascorbyl palmitate liposomes in the form of emulgel and cream. Tape stripping, as a minimally invasive technique, was applied to investigate the in vivo skin penetration of the examined formulations (LE, E, LC, and C) containing ascorbyl palmitate. This method provided an evaluation of the penetration profile and quantification of the amount of ascorbyl palmitate accumulated in the stratum corneum.

The overall percentages of ascorbyl palmitate extracted from all 16 tapes, compared to the starting amount of ascorbyl palmitate incorporated into the investigated formulations, are presented in Figure 2.

The results presented in Figure 2 indicate that the highest amount of ascorbyl palmitate was extracted from tape strips at sites where formulations with liposome dispersions were applied. A statistical comparison between the presented data revealed that there was a significant increase in the level of ascorbyl palmitate in the stratum corneum following the application of LE (93.31%) and LC (96.4%) when compared to both E (73.64%) and C (82.11%). The recovered amount of ascorbyl palmitate from LE and LC application sites after 120 min was significantly higher when compared with E and C application sites. The results showed that although there was an increase in the amount of penetrated L-ascorbyl palmitate in cream samples compared to emulgel samples (C compared to E, and LC compared to LE), this increase was not statistically significant. Therefore, it could be concluded that the incorporation into liposomes led to a statistically significant difference in the amount of penetrated active substance, regardless of the type of carrier used within the formulation.

The obtained results clearly indicated that encapsulation of ascorbyl palmitate promoted its penetration through the stratum corneum. The ability of liposomes to enhance the delivery of active ingredients from topical formulations has been attributed to their specific structure. Liposomes represent concentric bilayer vesicles that can fuse with other bilayers (cell membrane), releasing the content in this way [32]. Also, it is worth noting that the liposomes are loaded with active agents both inside and outside of their phospholipid membrane [54]. The presented results were in accordance with the data from the literature, which reported that topical application of liposomal formulations led to a significant increase in the rate and extent of drug absorption [58]. According to previous research, the hydrophobic liposome structure is responsible for the interaction with corneocytes to an extent that seems to be highly dependent on the lipid composition of liposomes. This interaction between lipid vesicles and the skin is also important to improve access to the epidermis of encapsulated active substances [61]. Contreras et al. have also studied hydroalcoholic gels that contain all-trans retinoic acid in the free and encapsulated forms in stratum corneum lipid liposomes, using the tape stripping method to establish the accumulation of the active substances on the surface and in the skin layers [62]. This method contributed to the overall conclusion that encapsulation of retinoic acid not only prolonged drug release but also promoted drug retention by the viable skin. Doi et al. investigated a serial tape stripping technique to detect the content of 3-O-cetyl ascorbic acid, one of the lipophilic L-ascorbic acid derivatives, in the stratum corneum, uptaken from examined cream [63].

During the tape stripping procedure, TEWL measurements were carried out before the application and detachment of the first adhesive tape, and then after 4, 8, 12, and 16 adhesive tapes were applied and removed successively from the same treated skin area. The purpose of the TEWL measuring was to detect the removal of the stratum corneum. Data from the literature suggest that an eightfold increase in TEWL indicates that the stratum corneum has been completely removed [35]. Based on the difference between basal TEWL values and measured values after 16 tape strips, it can be considered that the stratum corneum was removed completely from the application sites during our study (Figure 3).

After 12 tape strips in a uniform manner, more than 70% of the applied ascorbyl palmitate was found on the adhesive tapes from liposomal formulations (LE and LC), and about 60% from emulgels and creams that do not contain encapsulated L-ascorbic acid in the liposome dispersion (E and C) (Figure 4). The results presented in Figure 4 point out that ascorbyl palmitate had different penetration profiles for the different formulation types. The increase in TEWL corresponded to the depth of the penetration in the epidermis.

Texture and sensory analyses are generally employed in the pharmaceutical and cosmetic industry to assess the quality of manufactured products. As indicators of the mechanical and applicative properties of formulations, texture analysis serves as an objective method, while sensory analysis offers subjective insight. Together, they offer information that is important for both the activity of the formulations and the compliance of the consumers with the drug or cosmetic product [64].

The results of the TPA (Table 5) pointed out that the presence of liposome dispersion did not influence the adhesiveness of the emulgel. However, that was not the case with the cream. LC was the formulation with the highest adhesiveness, which indicates the stickiness of the product to other surfaces. On the other hand, the LC had the lowest cohesiveness of all tested samples. It indicates the strength of the internal bonds of the products. From these results, it can be concluded that dispersion caused the lowering of the cohesiveness since E and C had the greatest values of cohesiveness. Hardness describes the force required to rub a product between fingers and is inversely proportional to the spreadability of the preparation. The highest hardness values were recorded for the LE sample, while the lowest values were obtained for sample C. The values of hardness in cycle 2 were coherent with those of the first cycle. The lack of a significant difference between hardness cycle 1 and hardness cycle 2 values indicated that the structure of the preparation did not weaken after the first compression cycle.

The results of the sensory analysis were in accordance with the results of the texture analysis regarding consistency (Table 6). The sample LE was characterized with the highest consistency, which was expected based on the obtained results for the hardness parameter. The participants marked LE as the sample with the highest density during application, exhibiting the slowest absorption rate (Figure 5) and retaining the most expressed residual film (Figure 6). In addition, the participants marked sample LE as the one with the best spreadability, followed by LC and C, while E was characterized as the most difficult to spread.

The organoleptic properties of the formulations, summarized according to the results of the sensory analysis, favored the sample LE, since LE was described as the one with the highest gloss level before application and the one leaving the most gloss on the skin during the application.

Concerning sample LC, as another formulation with liposome dispersion, the texture analysis indicated that LC had the closest textural characteristics to LE. This was also observed for the sensory analysis.

By comparing the results of tape stripping, texture, and sensory analysis (Table 7), it was assumed that the penetration of the active substance—ascorbyl palmitate—in the skin from formulation LE, which was over 90% (93.31%), was due to its textural characteristics, in terms of best consistency and spreadability, when compared to the other tested formulations. The tape stripping results revealed that the amount of penetrated ascorbyl palmitate was even higher for LC (96.4%) (Figure 2). However, for TEWL values after 16 strip applications, LC and LE exhibited a 10-fold and 7-fold increase in relation to the initial values, respectively (Figure 4). This indicated that less of the stratum corneum was removed from the site where formulation LE was applied, explaining why the total amount of penetrated ascorbyl palmitate was lower for formulation LE in comparison to LC.

## 4. Conclusions

By examining the dermal penetration of liposomal and non-liposomal creams and emulgels with 2% ascorbyl palmitate using the tape stripping method and their sensory and applicative characteristics, the superiority of liposomal compared to non-liposomal samples was shown. Namely, liposomal encapsulation increased the amount of ascorbyl palmitate recovered in the stratum corneum when both liposome-based emulgel (LE) (93.31%) and cream (LC) (96.4%) were used as bases in comparison with the conventional emulgel (E) (73.64%) and cream (C) (82.11%). At the same time, it was shown that the use of cream as a base led to a better penetration profile compared to the use of emulgel (in the case of both liposomal and non-liposomal formulations).

Comparing the non-liposomal and liposomal formulations (cream and emulgel), liposomal cream and emulgel had good skin applicability (primarily spreadability) and sensory properties (the film formed on the surface of the skin creates conditions for a good moisturizing effect), while the non-liposomal formulation exhibited a weaker ability to penetrate and weak spreadability and did not leave a film on the surface of the skin (which can potentially reduce TEWL).

According to the current results, the liposomal emulgel revealed an enhanced penetration of ascorbyl palmitate reflected by the highest percentage of this substance recovered in the stratum corneum.

## Figures and Tables

**Figure 1 pharmaceutics-16-00962-f001:**
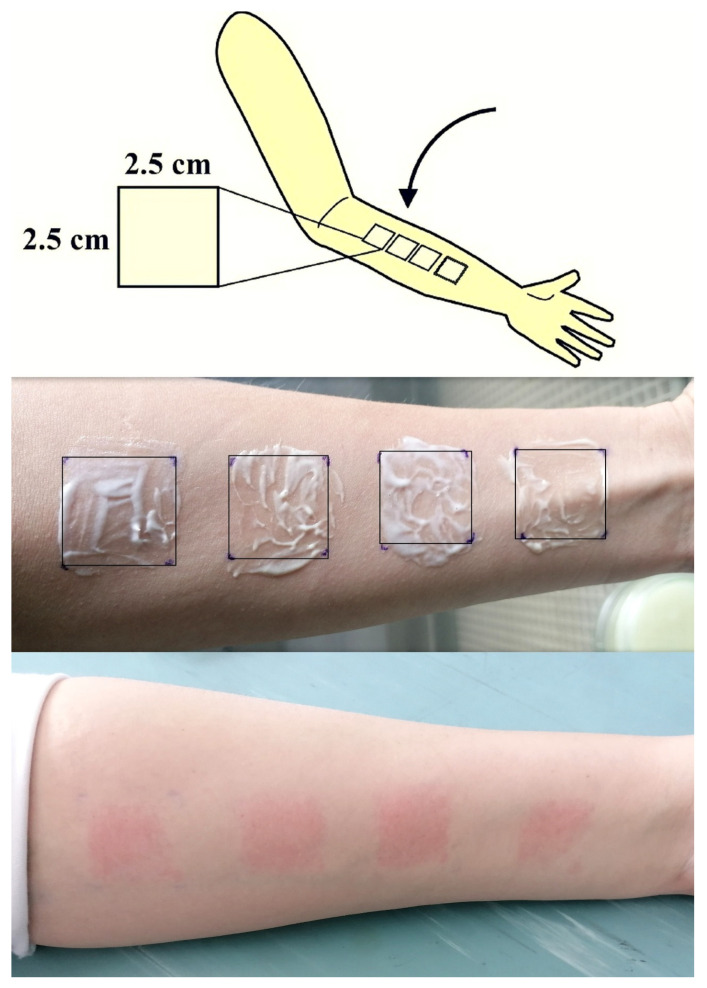
Tape stripping procedure.

**Figure 2 pharmaceutics-16-00962-f002:**
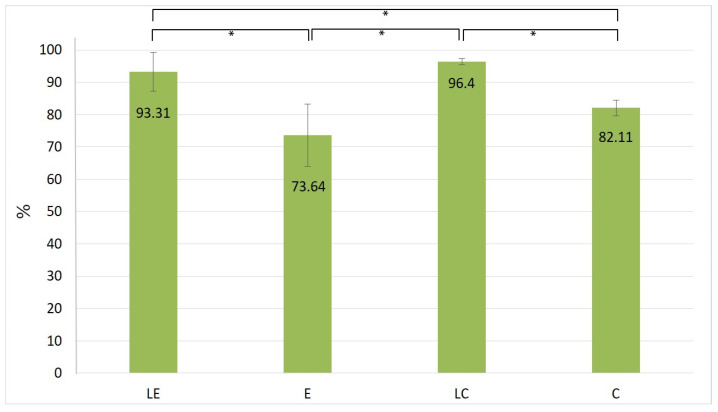
Total amount percentage of ascorbyl palmitate recovered in the stratum corneum for formulations LE, E, LC, and C after 2 h. Significant differences are marked with * (*p* < 0.05).

**Figure 3 pharmaceutics-16-00962-f003:**
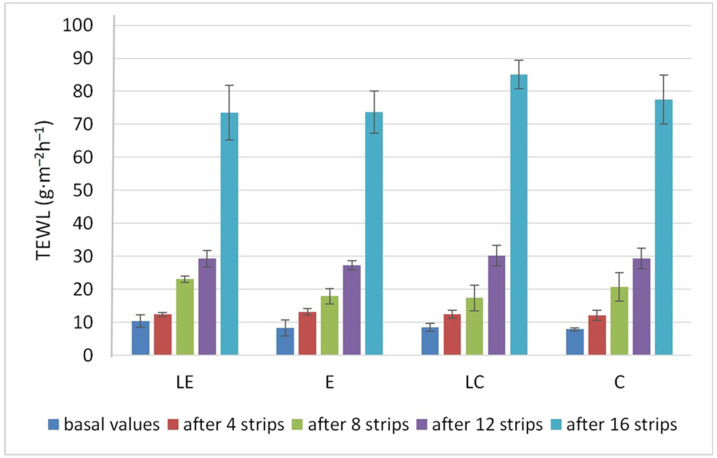
TEWL changes depending on the number of strips removed.

**Figure 4 pharmaceutics-16-00962-f004:**
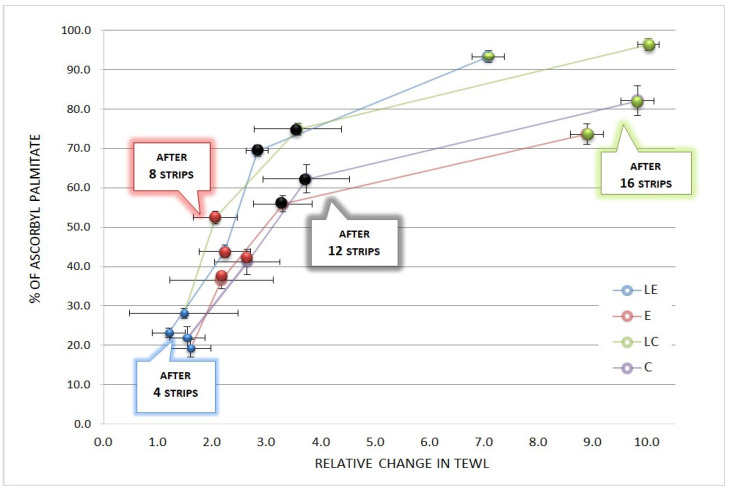
Percentage of the total amount of ascorbyl palmitate applied with each formulation (LE, E, LC, C) on the strips compared to the relative changes in TEWL after 4, 8, 12, and 16 tape strips were removed from the skin.

**Figure 5 pharmaceutics-16-00962-f005:**
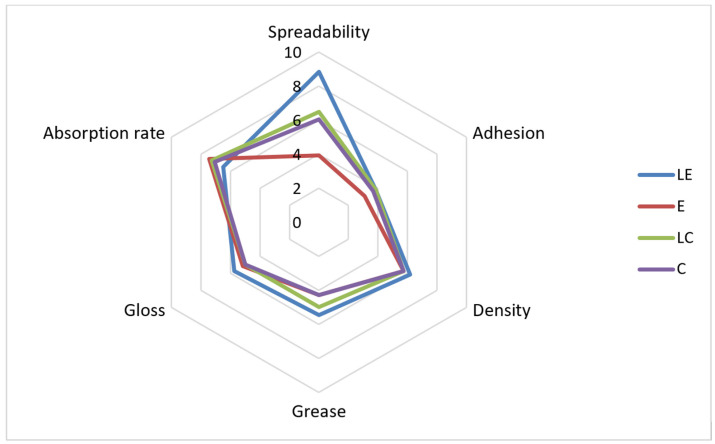
The results of sensory analysis of the investigated samples during application (LE, E, LC, C).

**Figure 6 pharmaceutics-16-00962-f006:**
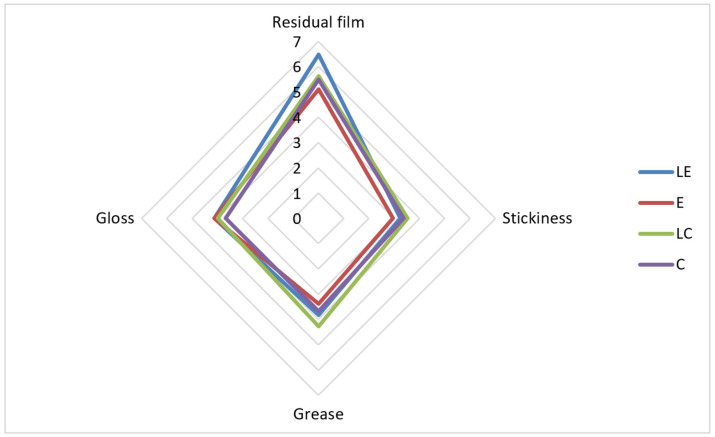
The results of sensory analysis of the investigated samples after application (LE, E, LC, C).

**Table 1 pharmaceutics-16-00962-t001:** Quantitative composition (%, (*w*/*w*)) of ascorbyl palmitate dispersion.

Ingredients (INCI Name)	%, (*w*/*w*)
Phosal 40IP	10.00
Ascorbyl palmitate	5.00
Propylene glycol	10.00
Phenoxyethanol (and) Ethylhexylglycerin	1.00
Aqua (Water)	ad 100.00

**Table 2 pharmaceutics-16-00962-t002:** Qualitative and quantitative compositions (%, (*w*/*w*)) of investigated formulations.

Ingredients (INCI Name)	LipoEmulgel (LE)	Emulgel (E)	LipoCream (LC)	Cream (C)	Function in the Formulation
**Oil Phase**					
Caprylic/capric triglycerides	11.00	11.00	11.00	11.00	Emollient
Isopropyl myristate	7.50	7.50	7.50	7.50	Emollient
Olive oil	3.00	3.00	3.00	3.00	Emollient
Cetearyl alcohol (and) Coco-glucoside	7.00	7.00	7.00	7.00	O/W emulsifier
Myristyl alcohol (and) Myristyl glucoside	1.50	1.50	1.50	1.50	O/W emulsifier
Ascorbylpalmitate	-	2.00	-	2.00	Active substance
Ascorbylpalmitatedispersion	40.00	-	40.00	-	Active substance
**Water phase**					
Hydroxyethyl cellulose (HEC)	1.00	1.00	-	-	Thickener/ Gelling agent
Propylene glycol	10.00	10.00	10.00	10.00	Humectant
Phenoxyethanol (and) Ethylhexylglycerin	1.00	1.00	1.00	1.00	Preservative
Aqua (Water)	ad 100.00	ad 100.00	ad 100.00	ad 100.00	Solvent

**Table 3 pharmaceutics-16-00962-t003:** The conditions for texture profile analysis (TPA).

Parameter	Value
**Test Speed**	2 mm/s
**Target Value**	2 mm
**Trigger load**	10 g
**Probe**	Cone probe, TA-STF
**Measured parameters**	Hardness cycle 1 Hardness cycle 2 Cohesiveness Adhesiveness

The TPA was performed in triplicate.

**Table 4 pharmaceutics-16-00962-t004:** pH, electrical conductivity (μS/cm) values, and organoleptic properties of LE, E, LC, and C samples before and after centrifuge assay, as well as after accelerated and long-term stability test.

**pH**				
	**Before Centrifuge Assay**	**After Centrifuge Assay**	**After Accelerated** **Stability Test**	**After 30 Days** **(21 ± 2 °C)**
**LE**	4.50	4.55	4.51	4.53
**E**	4.66	4.65	4.59	4.56
**LC**	4.39	4.31	4.35	4.35
**C**	4.90	4.93	4.91	4.87
**Electrical Conductivity (μS/cm)**				
	**Before Centrifuge Assay**	**After Centrifuge Assay**	**After Accelerated** **Stability Test**	**After 30 Days** **(21 ± 2 °C)**
**LE**	50.20	51.10	51.44	51.41
**E**	52.90	50.40	52.95	52.47
**LC**	39.60	37.80	40.11	39.87
**C**	59.10	57.10	56.15	58.65
**Organoleptic Properties (Color, Smell, Appearance)**				
**LE**	yellowish-white, no odor, glossy	yellowish-white, no odor, glossy	yellowish-white, no odor, glossy	yellowish-white, no odor, glossy
**E**	white, no odor, glossy	white, no odor, glossy	white, no odor, glossy	white, no odor, glossy
**LC**	yellowish-white, no odor, glossy	yellowish-white, no odor, glossy	yellowish-white, no odor, glossy	yellowish-white, no odor, glossy
**C**	yellowish-white, no odor, glossy	yellowish-white, no odor, glossy	yellowish-white, no odor, glossy	yellowish-white, no odor, glossy

**Table 5 pharmaceutics-16-00962-t005:** Results of the texture profile analysis of the formulations LE, E, LC, and C.

	Adhesiveness (mJ)	Cohesiveness	Hardness Cycle 1 (g)	Hardness Cycle 2 (g)
LE	0.43 ± 0.06	1.54 ± 0.18	27.67 ± 3.79	25.67 ± 4.04
E	0.43 ± 0.06	1.78 ± 0.15	25.33 ± 1.53	24.33 ± 1.53
LC	0.50 ± 0.20	1.48 ± 0.28	25.67 ± 4.51	24.33 ± 5.03
C	0.33 ± 0.06	1.74 ± 0.24	23.33 ± 2.52	22.00 ± 2.65

**Table 6 pharmaceutics-16-00962-t006:** Results of the sensory analysis of the formulations LE, E, LC, and C before the application.

Before Application				
	LE	E	LC	C
Consistency	10.00	9.71	9.71	9.71
Gloss level	6.95	6.12	6.59	6.12

**Table 7 pharmaceutics-16-00962-t007:** Comparison of the characteristics of the samples LE, E, LC, and C. The sample with the most pronounced characteristic is marked with “+”.

		LE	E	LC	C
**Physico-chemical characteristics)**	Organoleptic properties	Acceptable	Acceptable	Acceptable	Acceptable
	pH	Within the range suggested for topical preparations	Within the range suggested for topical preparations	Within the range suggested for topical preparations	Within the range suggested for topical preparations
**Tape stripping**	Total percentage of penetrated ascorbyl palmitate	>90%	<90%	>90%	<90%
				+	
**Sensory analysis**	Consistency	+			
	Gloss	+			
	Spreadability	+			
	Residual film	+			
	Fast absorption		+	+	+
	Slow absorption	+			
	The least sticky		+		
	The least greasy feeling on the skin		+		+
**Texture analysis**	Hardness	+			
	Consistency	+			
	Cohesiveness		+		+
	Adhesiveness			+	
	Spreadability				+
	Deformity after pressure	Stable structure	Stable structure	Stable structure	Stable structure

## Data Availability

The data presented in this study are contained in the article.

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
