# Peer review of "Liposomal Encapsulation of Ascorbyl Palmitate: Influence on Skin Performance"

_pharmaceutics, 2024, doi:10.3390/pharmaceutics16070962_

Round 1
Reviewer 1 Report
Comments and Suggestions for Authors
The manuscript entitled "Enhanced skin penetration of ascorbyl palmitate achieved via liposomal encapsulation: tape stripping evaluation" is designed and written well. It needs some edits prior to acceptance.
In abstract, line#27, replace 'a better penetration' with statistical significance level comparison. Write the folds improvement from bot products.
2.2.1, line#106 - used to prepare liposomal formulation. Please check the statement and correct it as liposomal topical formulations.
Table 1 - check the legend why term qualitative used in the legend.
Write a comparative analysis of previously reported studies with liposomal creams and emulgels.
Write the evidence or supportive data for 5%w/w ascorbyl plamitate used for the liposomal formulation.
stability of the lead formulations should be include in the manuscript.
What could be the reasons for changes in relative TEWL with %ascorbyl palmitate. Please explain?
Comments on the Quality of English Language
Only minor edits needed.
Reviewer 2 Report
Comments and Suggestions for Authors
The manuscript entitled “Enhanced skin penetration of ascorbyl palmitate achieved via liposomal encapsulation: tape stripping evaluation”, written by Aleksandra Stolić Jovanović et al., covers the interesting topic of dernal delivery of lipophilic derivative of vitamin C, i.e. ascorbyl palmitate. The penetration ability of ascorbyl palmitate incorporated conventional semi-solid dosage forms as creams and emulgels and liposomes used as nanocarriers was compared with regard to kinetics of penetration of actives, studied by tape stripping method in vivo. Parallelly, sensorial analysis of tested formulations was also performed in addition to textural analysis. The manuscript is well written and the data are well structured, being of particularly relevance for readers involved in dermo-cosmetic science and dermal delivery of active ingredients. On my opinion the manuscript is suitable for publishing in Pharmaceutics after some revision. My specific comments to the authors are listed below:
- For better comprehensibility of the paper I would suggest the authors to explain the term emulgel – are this semi-solid emulsion or semi-solid nanoemulsion? It is written that they are obtained by conversion of classic emulsions by adding a gelling agent in the water phase, resulting in transparent vehicles. Nevertheless, classic emulsions are defined as coarse dispersions, therefore their appearance is milky also upon thickening. So, I would suggest to discuss this a bit.
- In the introduction part it would be useful to add a short overview of liposomal and classical formulations in addition to other semi solid formulations with ascorbyl palmitate previously studied (focusing on stability and dermal absorption of ascorbyl palmitate).
- In the method section, the method used for determine the encapsulation efficacy should be explained more precisely. Was it tested by dialysis bag technique?
- Line 174: it is written that “The literature data suggests that an eightfold increase in TEWL indicates that stratum corneum was completely removed [30]”. I agree, nevertheless, the literature data also suggest that the skin barrier is disrupted when the TEWL values are above 25 g/m2/h.
- Line 195, was stability of ascorbyl palmitate tested during storage of samples in a freezer prior HPLC analysis?
- As the liposomes size and polydispersity index are rather high (0.67±0.01), I am wondering why the authors didn`t decide to perform additional extrusion cycle?
- What was the dilution factor prior the size measurement by Zetasizer? It is common to dilute the liposomal dispersion between 10- to 100-fold to avoid multiple scattering, so maybe this could be the reason for high PDI. This could be easily tested.
- Line 323, the authors mention the different kinetics of ascorbyl palmitate absorption from tested systems, but no kinetics was calculated, though. So, I would suggest to adjust the terminology.
